# Implicit water model within the Zimm-Bragg approach to analyze experimental data for heat and cold denaturation of proteins

Artem Badasyan [1✉], Shushanik Tonoyan[2], Matjaz Valant [1,3] & Joze Grdadolnik [4]

Studies of biopolymer conformations essentially rely on theoretical models that are routinely used to process and analyze experimental data. While modern experiments allow study of single molecules in vivo, corresponding theories date back to the early 1950s and require an essential update to include the recent significant progress in the description of water. The Hamiltonian formulation of the Zimm-Bragg model we propose includes a simplified, yet explicit model of water-polypeptide interactions that transforms into the equivalent implicit description after performing the summation of solvent degrees of freedom in the partition function. Here we show that our model fits very well to the circular dichroism experimental data for both heat and cold denaturation and provides the energies of inter- and intra-molecular H-bonds, unavailable with other processing methods. The revealed delicate balance between these energies determines the conditions for the existence of cold denaturation and thus clarifies its absence in some proteins.

[1] University of Nova Gorica, Materials Research Laboratory, Nova Gorica, Slovenia. [2] Yerevan State University, Department of Molecular Physics, Yerevan, Armenia. [3] University of Electronic Science and Technology of China, Institute of Fundamental and Frontier Sciences, Chengdu, China. [4] National Institute of Chemistry, Ljubljana, Slovenia. ✉email: abadasyan@gmail.com

 COMMUNICATIONS CHEMISTRY | https://doi.org/10.1038/s42004-021-00499-x

To become biologically active, most proteins fold to globular conformations that are stabilized by many different non-covalent interactions. Among all non-covalent interactions, hydrogen bonding plays a crucial role in protein folding process. It is therefore extremely important to understand the nature of the interactions between the protein constituents and their interaction with the surrounding water molecules and to estimate the strength of these interactions. The (un)folding of proteins is one of the most studied phenomena in the field of Biophysics[1]. The final result of a (un)folding experiment is usually represented by a sigmoidal curve of the order parameter (degree of nativeness, helicity degree, etc.) versus an external variable (temperature, pressure, pH, etc.)[2]. The order parameter is expressed through the first derivative of thermodynamic potential, while the second derivative is related to the heat capacity. Once the expression for thermodynamic potential (usually Gibbs free energy) is accepted, a theoretical expression is derived and fitted to experimental data points in order to estimate the (temperature-independent) enthalpic, entropic, and heat capacity costs of (un)folding (ref. [3] and references therein). Despite the growing interest in protein folding and protein structures, quantitative estimates of the strength of hydrogen-bonding interactions are available on the basis of experiments mainly for model compounds[4,5] or are calculated at different levels of theory[6,7]. While experimental methods have reached an unprecedented level and enable protein folding studies in vivo[8], theoretical models used in the fitting procedure have remained the same since the mid-20th century. Experimental data are usually processed either with the help of the two-state[9] or the Zimm–Bragg[10] approach. Each of these phenomenological thermodynamical theories has its benefits and drawbacks, but the most relevant is that they are both weak in describing the interactions between the polypeptide chain and the water molecules that are essential for proteins. Since it is generally accepted that the hydrophobic effect is the main driving force of folding[11,12], we conclude that the process of information extraction from folding experiments suffers from a lack of a theoretical description of water–protein interactions, especially when both cold (re-entrant) and heat denaturation are observed. Meanwhile, the standard two-state approach considers protein denaturation as a quasi-chemical reaction between the native and denatured states and relies on constant heat capacity ansatz[13,14]. Succeeded by Taylor expansions of enthalpy and entropy, truncated at second term[13–16], it leads to a quadratic dependence of Gibbs free energy on temperature or pressure, a formula similar to the one proposed by Hawley about half a century ago[17].

An interesting paper by Dill et al., devoted to the investigation of the hydrophobic effect, have explained the cold denaturation as appearing due to the weakening of the interactions with the solvent[18]. Their thermodynamic mean-field theory widely uses Flory-Huggins-like approach with the effective energy of solvophobic interaction, defined by the Hawley[17] formula, and reaches the qualitative agreement with some calorimetric experiments.

Although the two-state formulas usually fit the experimental points well, they are known to result in non-matching sets of fitted parameter values for circular dichroism (CD) and calorimetric data of the same protein[3]. In addition, the limits of applicability of the expansion-based and constant heat capacity approximation are not clear. And after all, the most important: in the picture of protein folding as drawn by the two-state model there are no parameters of the water–protein interactions. With the approach we suggest, a procedure for the calorimetric data can be formulated as well. It will be interesting to see if it gives a similar set of fitted parameters, however, we will leave this

question for future studies and concentrate here on introducing the method to extract the parameters of the water–protein interactions from the circular dichroism data.

To introduce water into the protein folding story, we have to suggest some simplifications, since water is the trickiest known solvent. First, in the physiologically relevant range and close to normal conditions, water has no phase transitions; second, the water–polypeptide interactions perturb only the first water layer[19]; and third, the experiments are carried out in a differential scheme (pure solvent signal is subtracted). Thus, we do not need to describe the behavior of bulk water per se: instead, it is sufficient to have a simple but qualitatively correct model capable of describing the short-range water–polypeptide interactions in the physiological temperature and pressure range, far from phase transitions of water.

Recently, analytical descriptions of polypeptide conformations in water, that lead to cold denaturation have been suggested[20–29]. To describe the water–polypeptide interactions, all of these authors relied on the multi-valued Potts spin models, which are best suited for directional interactions, such as H-bonding. Although qualitatively successful, none of these approaches provide a practical answer to the question: How to extract the constants of the interaction with water from the experimental data?

The Zimm–Bragg model[10] formulated in the 1950s, is so successful in describing conformational transitions in biopolymers that it is still widely used to treat experimental data[30–32]. In addition to its strengths, the original model formulation (see Supplementary Methods) also contains essential weaknesses, mostly due to the absence of the microscopic Hamiltonian. Indeed, it is not clear how the parameters $s$ and $\sigma$ should be amended to describe solvent effects. A better strategy we have been pursuing is to introduce the interactions with water into the Hamiltonian, further construct the corresponding transfer-matrix, and derive the characteristic equation of the model. The comparison with the original characteristic equation of the Zimm–Bragg approach will make it clear which model parameters are influenced by the presence of water in the model.

In line with the recent observation that the Zimm–Bragg sequential unfolding picture provides a better description of the heat denaturation experiment compared to the two-state approach[3], we have significantly improved the theoretical support of experimental data processing by explicitly introducing the water–polypeptide interactions into the Zimm–Bragg Hamiltonian. The proposed upgrade of Zimm–Bragg model has been tested to both heat and cold denaturation phenomena and allows to extract the energetic and entropic costs of H-bonding.

## Results and discussion

### The presence of water transforms the stability parameter $s$ of the Zimm–Bragg approach.

We start from the Potts-like spin Hamiltonian formulation of the Zimm–Bragg model[33]. Coupling with the approach used to describe solvent-biopolymer interactions[21,26,27] enables us to achieve our goal of a simple and analytically treatable model. Summing out the solvent degrees of freedom in the partition function results in the transformed stability parameter $s$ (see "Methods", Eq. (14)).

Although simple, the transformation qualitatively changes the behavior of the stability parameter of the Zimm–Bragg model as shown in Fig. 1. It happens because Eq. (14) actually encodes the water-related effects into the functional dependence of $\widetilde{s}$ on its parameters. For convenience, we choose internal units of temperature as $\tau = T \ln Q/U$ to put the transition temperature of water-free model transition at $\tau = 1$. Energetic parameters can be grouped as $\alpha = \frac{E-U}{U}$, and the reference water-free model

corresponds to $\alpha = -1$. Then,

$$\widetilde{s}(\tau, \alpha, Q, q) = \frac{1}{Q}\left[\frac{q^2 e^{1/\tau}}{(q - 1 + e^{\frac{1+\alpha}{2\tau}})^2} - 1\right].\qquad(1)$$

Figure 1 reveals the waterfall-like behavior of $z = \widetilde{s}(\tau, \alpha)$ function. $z = 1$ surface separates the native conformations from disordered. The intercept of these two surfaces determines the onset of conformational transitions in the system. However, these waters are muddy; there is an important non-monotonicity hidden behind the waterfall.

To make it more evident, we analyze the projections. In Fig. 2a starting from the $\alpha = -1$ (no water, $E = 0$), in the $\alpha < 0$ ($E < U$) region the curve is monotonic. At $\alpha > 0$ ($E > U$) the maximum appears and the $\alpha = 0$ ($E = U$) value separates two regimes. The intercept with $\widetilde{s} = 1$ graphically determines transition temperature. For $\alpha > 0$ there are either two (corresponding to cold and hot denaturations) or no intercepts, depending on the height of the maximum (which has to be > 1). Physically, it means that only at solvent-protein energies $E$ slightly above (in absolute units) the intra-protein interaction energies $U$, both cold and heat denaturations take place. At larger energies $E$ there is no transition, and the system is denatured at any temperature. If the denaturant (water, in this case) is strong enough to break

internal H-bonds, the protein will remain disordered at any temperature. Figure 2b shows the energy dependence of $\widetilde{s}$ at different temperatures, which becomes non-bijective for $\alpha > 0$. $\tau = 1$ is the highest transition temperature possible, and it takes place at $\alpha = -1$ case of no water. At all other energies, transitions temperatures are lower than for in vacuo case. The growth of $\alpha$ moves the intercept with the straight line at 1 to lower temperatures. Above the $\alpha = 0$ value the $\widetilde{s}(\alpha)$ function is not bijective anymore: to every value of $\alpha$ there are two values of $\widetilde{s}$. It is obvious that the situation of two conformational transitions can be reached by smooth changes in energy balance. However, the inconvenience of operating with non-bijective function is also clear. The last mathematical fact may lead to serious consequences, when the protein is denatured by changing the denaturant concentration, and its order content is detected at room temperature. It would be safer to scan over the accessible temperature range instead.

**Fitting formulas**. The formulas of the previous section are written using the units that are convenient for theoretical analysis, with some of the constants set to unity and the temperature in energy units. Since our purpose here is to provide formulas that can be fitted to experimental data, conversion to laboratory units is required, along with other adjustments. Due to the presence of water, the energetic cost of H-bonding between N−H and O=C groups must be understood as $U = U_{pp} + U_{ss}$, where $U_{pp}$ and $U_{ss}$ are the energies of the intramolecular H-bonds and solvent-solvent H-bonds, respectively. To be able to correctly estimate the balance of energies in our simplified model for water, we have to take into account, that to be able to form a H-bond with N−H or C=O groups, the water–water bond has to be broken. Besides this point, we do not describe the water–water interactions in our simplified approach to water–protein interactions. It is convenient to introduce

$$h = U/2 = (U_{pp} + U_{ss})/2; \quad h_{ps} = E = U_{ps},\qquad(2)$$

where $h$ represents a single H-bond energy within the polypeptide, and $h_{ps}$ corresponds to a single polypeptide-solvent H-bond.

Recently, theoretical studies of hydrated proteins report the need for introducing the temperature shift to reflect the super-Arrhenius relaxation observed in the experiment. Thus, $t - t_0$ temperature shift appears in hydrated proteins, because of the presence of partially glassy states; $t_0$ is a fitting parameter, that represents the glass transition temperature in supercooled liquid (see "Methods").

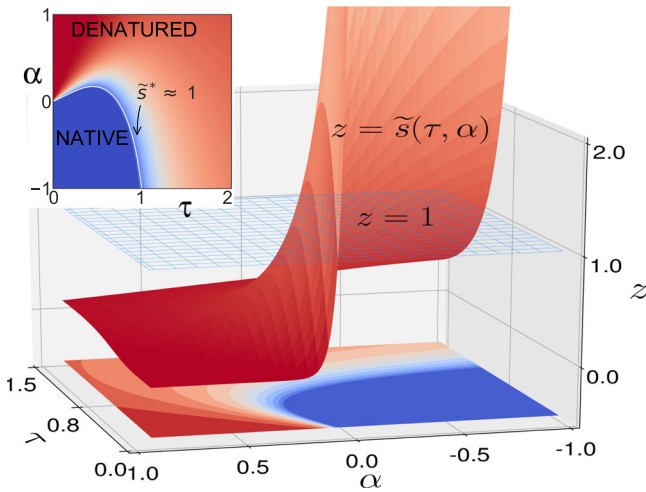

**Fig. 1 Water(fall) in the Zimm–Bragg model.** 3D graph of re-defined stability parameter $\widetilde{s}(\tau, \alpha)$ from Eq. (1). Inset: the phase diagram, resulting from the intercept between the surfaces; $\widetilde{s} \approx 1$ separates the native conformations from the denatured. Temperature in $\tau = \frac{T \ln Q}{U}$ units and energy as $\alpha = \frac{E-U}{U}$.

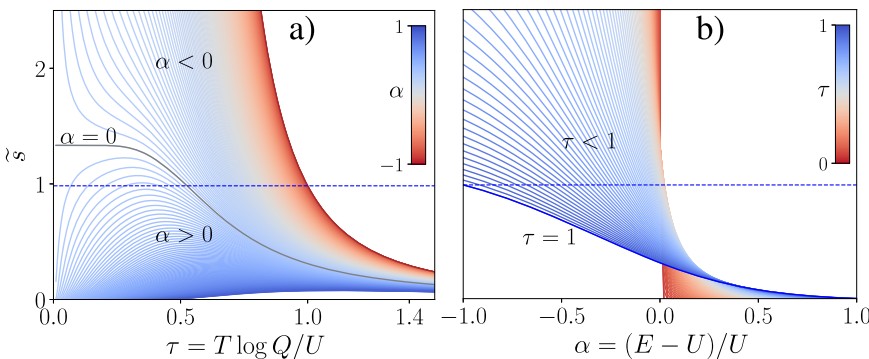

**Fig. 2 Renormalized stability parameter.** Projections of $\widetilde{s}(\tau, \alpha)$ at fixed values of one of parameters, shown on the colorbar. **a** $\widetilde{s}(\tau)$ at different $\alpha$s (temperature dependence at different energies). **b** $\widetilde{s}(\alpha)$ at different $\tau$s (energetic dependence at different temperatures).

After the substitutions, Eq. (1) will become

$$\widetilde{s}(t, t_0, h, h_{ps}, Q, q) =$$

$$\frac{1}{Q}\left[\left(e^{-h/(t-t_0)} + \frac{e^{(h_{ps}-h)/(t-t_0)} - e^{-h/(t-t_0)}}{q}\right)^{-2} - 1\right], \quad (3)$$

where $t = RT$, $R$ is the gas constant, and other parameters are as described above. In addition, the value of the entropic cost of a H-bond with water is set to $q = 16$ (see "Methods" for details). Equation (3) allows for both monotonic and nonmonotonic behaviors for the Zimm–Bragg stability parameter in water, as can be seen in Fig. 2a. The insertion of $\widetilde{W}$ into Eq. (14) and other consequent expressions up to the order parameter leads to the formula we can finally fit to experimental data (see Supporting Info).

**Experimental data analysis**. The resulting order parameter is a function of temperature (see Eq. (16) for the explicit expression)

$$\theta(\widetilde{s}, \sigma) = \theta(t; \underline{t_0, h, h_{ps}, Q}), \quad (4)$$

and contains four fitting parameters (underlined): $t_0$, the temperature shift value; $h$, the energy of intramolecular H-bonds; $h_{ps}$, the energy of the intermolecular polypeptide-solvent H-bond, and the entropic cost of a H-bond $Q$, related to the cooperativity parameter by $\sigma = 1/Q$. We do not exclude the possibility to determine some of these parameters from the independent measurements or simulations, but this question is out of the scopes of the presented study.

To demonstrate the efficiency of the proposed method, we have selected data sets from four published studies: Go et al.[34], Seelig et al.[3], Aznauryan et al.[35], and Bryson et al.[36]. The experimental data sets show the temperature dependence of order parameter measured by different methods, different experimental arrangements, and different groups from the late 1960s to 2010s.

*Case 1.* Go et al. (Fig. 3a) study[34] is one of the first attempts to include water–polypeptide interactions in the Zimm–Bragg approach, which allowed to describe the experimental data for poly-glycine or poly-L-alanine in water. In particular, the data set from Figure 3 of Go et al.[34] shows the temperature behavior of the helicity degree of poly-L-alanine chains of different lengths in water. The fit, presented in their Fig. 3[34] is obviously very poor (see Supplementary Discussion), as a consequence of the quadratic approximation[15,17] used by Go et al.[34]. The fitted curves are non-monotonous and indicate the presence of cold denaturation. This contradicts the experimental data in their Fig. 3[34]. To qualitatively improve the description, we use our Eq. (4) to fit the same data set. It results in a nicely fitted curve (Fig. 3a) with reasonable values for the H-bond energies about 5–6 kJ mol⁻¹ (Table 1a). Although the theoretical approach we propose always takes into account the presence of water (as in the experiment), it does not necessarily lead to the non-monotonous order parameter and cold denaturation (again, exactly as it is in the experiment). We thus show a more adequate description of the thermal behavior of the system, as compared to Go et al.[34].

*Case 2.* Seelig et al.[3] (Fig. 3b) provide a solid and valuable example of a standard approach to the problem. They have compared the application of the two-state approach and Zimm–Bragg theory in its simplest formulation for the processing of several experimental data sets on protein folding. They concluded that the Zimm–Bragg approach, augmented by the Hawley-like free energy[13,14,17], describes the protein folding experiments better than the two-state approach. As can be seen in Fig. 3b, our Eq. (4) from the Hamiltonian-base approach describes recent CD experimental data[3] very well (although, due to the small number of experimental points, the fitting error is large; see Table 1b).

*Case 3.* Aznauryan et al.[35] (Fig. 3c) have investigated the temperature dependence of the cold- and heat-denatured states of yeast frataxin (Yfh1) by means of CD and Forster resonance energy transfer experiments. These authors have also used a two-state model with Hawley-like free energy[13,14,17]. Since Aznauryan et al.[35] deliberately chose a protein that undergoes both cold and heat denaturation in the physiological range of the external conditions, their data is optimal in view of testing our theoretical curve by fit. As before the fit in Fig. 3c is very good, the fitting errors are the smallest and very reasonable energies of the H-bonds at about 1.42 kJ mol⁻¹ (Table 1c) are obtained.

*Case 4.* Bryson et al.[36] (Fig. 3d) have reported on the thermal denaturation of α3C de novo designed three-helix-bandle protein, measured with far-UV CD, additionally treated with different concentrations of the guanidinium hydrochloride *GdnHCl* denaturant. Based on the two-state approach, the authors succeeded in reproducing experimentally detected conformational transitions (see Fig. 6[36]), however, no cold denaturation was observed at a zero denaturant concentration in the range of temperatures between the two (solid–liquid and liquid–vapor) phase transitions of water. At a denaturant concentration of 1–2 M, heat denaturation is shifted toward lower temperatures and the pattern of experimental points shows the signs of non-monotonic behavior for 2 M solution at the lowest temperatures. At 2.5–3 M of *GdnHCl*, the curve clearly loses monotonic behavior and cold denaturation appears in addition to heat denaturation. As shown in Fig. 3d, our formula Eq. (4) is able to describe this complicated case as well. According to the definition of parameter $\alpha$, this is a clear indication of the increased prevailance of intramolecular H-bonds over the intermolecular ones, or strengthening of intramolecular H-bonds. The increase of denaturant concentration (from the upper curve to lower) is mimicked by the increased value of $\alpha$. It corresponds to an absolutely transparent and logical physical picture of altered balance between inter- and intramolecular H-bonds due to the enhanced role of denaturant. The resulting values for the H-bond energies are in the range of 1.2–2.5 kJ mol⁻¹ (Table 1d).

It is informative to see the temperature behavior of $\widetilde{s}(T)$ at different denaturant concentrations, mimicked by the different fitted $\alpha$s (Fig. 4). The data of Bryson et al. at 1 M of *GdnHCl* indicate a monotonic temperature behavior at $\alpha < 0$ with the presence of only the heat denaturation. Yet at the other four (higher) concentrations of *GdnHCl* resulted in $\alpha > 0$, $\widetilde{s}$ becomes a curve with the maximum, indicating the presence of both heat and re-entrant, cold denaturation. The data of Bryson et al. enabled illustrating the appearance of previously absent cold denaturation as a result of the shifted balance between interaction energies in the system.

## Conclusion

In the cases considered above, we have shown the validity of the approach in describing conformational reorganizations in poly-L-alanine[34] homopolypeptide, natural recombinant human Apo A-1 protein, and lysozyme[3], intrinsically disordered yeast frataxin (Yfh1)[35], as well as α3C de novo designed three-helix-bandle protein[36]. As a result of the fit, we enable the access to the bonding energies with water, information that is always present in CD spectroscopic data but has been unavailable up to now.

From the above results, we conclude that the least-square fit of the proposed Eq. (4) leads to reasonable values for all parameters in all four cases with the coefficient of determination $R^2 > 0.99$

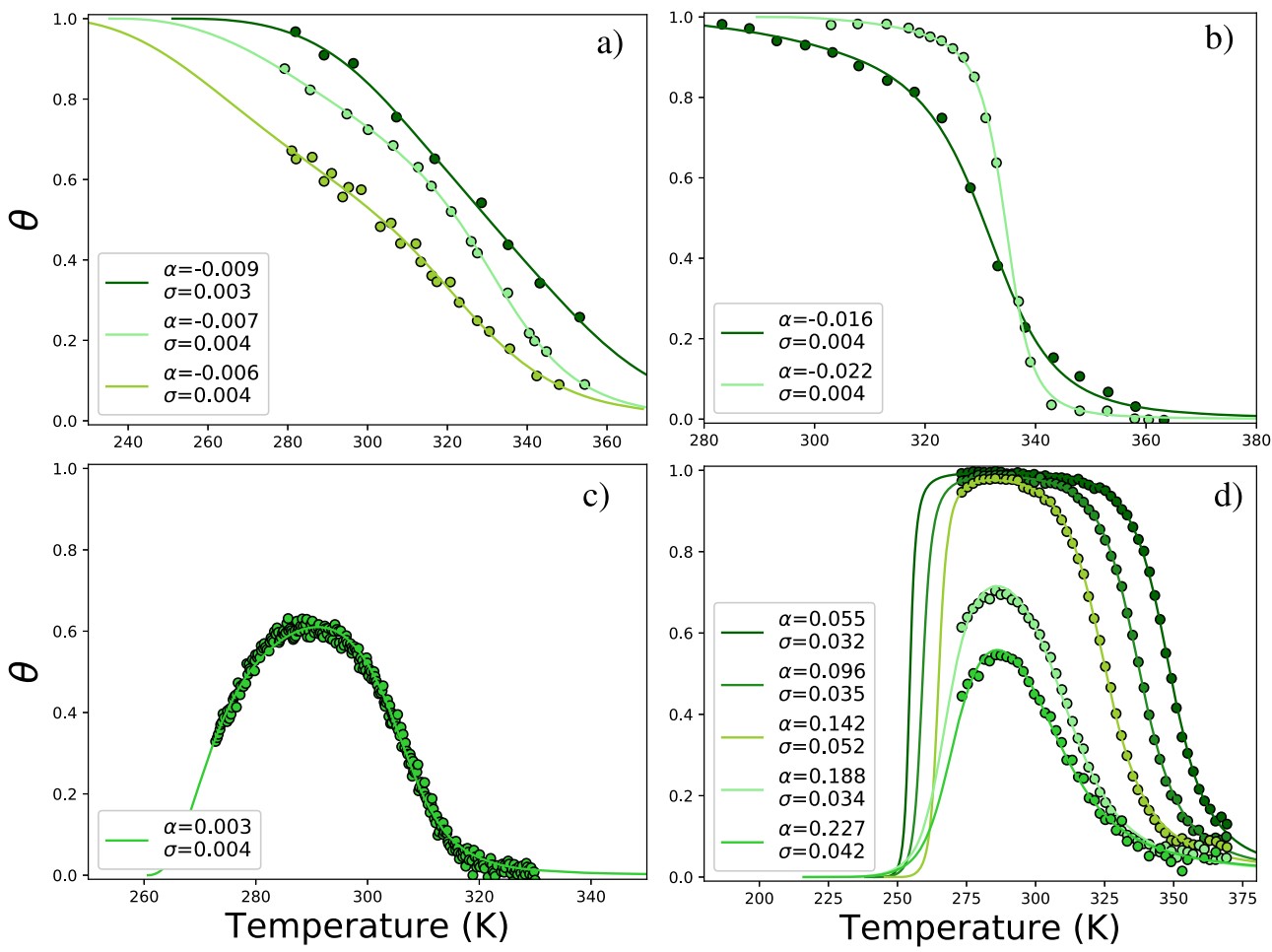

**Fig. 3 Fitted order parameter curves.** Results of fitting Eq. (4) to experimental points. Recalculated values of the cooperativity (entropic) parameter $\sigma = 1/Q$ and the energetic parameter $\alpha = \frac{2U_{ps} - (U_{pp} + U_{ss})}{U_{pp} + U_{ss}} = \frac{h_{ps} - h}{h}$ are shown in the figures. Data are taken from **a** Go et al.[34], **b** Seelig et al.[3], **c** Aznauryan et al.[35], **d** Bryson et al.[36].

| Table 1 Fit results with errors in brackets (percents). | | | | | | |
|---|---|---|---|---|---|---|
| | $t_0$, K | $h$, J mol$^{-1}$ | $h_{ps}$, J mol$^{-1}$ | $Q$ | $\sigma$ | $R^2$ |
| (a) Data from Fig. 3 of Go et al.[34] | | | | | | |
| Darkgreen | 251.0(4.8) | 5754(16.1) | 5701(15.9) | 299(4.3) | 0.003 | 0.997 |
| Lightgreen | 235.2(2.2) | 5086(4.7) | 5049(4.66) | 275(0.8) | 0.004 | 0.999 |
| Yellowgreen | 216.4(8.6) | 5761(14.4) | 5727(14.3) | 278(1.47) | 0.004 | 0.992 |
| (b) Data from Figs. 4 and 6 of Seelig et al.[3] | | | | | | |
| Darkgreen | 247.6(5.62) | 4018(13.6) | 3955(13.3) | 279(5.2) | 0.004 | 0.998 |
| Lightgreen | 289.4(2.5) | 1973(11.3) | 1930(11.0) | 273(10.2) | 0.004 | 0.999 |
| (c) Data from Fig. 1C of Aznauryan et al.[35] | | | | | | |
| Darkgreen | 260.6(0.42) | 2134(1.8) | 2141(1.9) | 233(0.7) | 0.004 | 0.994 |
| (d) Data from Fig. 6 of Bryson et al.[36] | | | | | | |
| Darkgreen | 241.2(2.4) | 1954(8.3) | 2068(7.6) | 32(9.4) | 0.032 | 0.999 |
| Forestgreen | 238.2(1.6) | 1792(6.7) | 1981(6.28) | 28(8.0) | 0.035 | 0.999 |
| Yellowgreen | 245.3(1.0) | 1267(5.7) | 1476(5.5) | 19(6.5) | 0.052 | 0.999 |
| Lightgreen | 215.9(1.6) | 1971(8.2) | 2428(6.7) | 29(11.8) | 0.034 | 0.997 |
| Limegreen | 215.9(2.1) | 1771(11.8) | 2292(9.3) | 24(16.0) | 0.042 | 0.993 |

(see Table 1). All the fitted results indicate a delicate balance between the intra- and intermolecular energies, resulting in the appearance of cold denaturation. The energies of the peptide-water H-bonds in Eq. (2) correlate with the energy obtained from the literature, although direct comparisons are quite difficult. At the same time, we are aware that there are several aspects that are known to influence H-bonding but they are not captured in the proposed approach. As was shown by Avbelj and Moult, peptide solvation is affected by the electrostatic screening[37], responsible for H-bond strengthening in the vicinity of polypeptides and possibly related to the cooperative effects in water H-bonds[38,39]. Although the gas–liquid model reports higher enthalpies[40–42], the H-bond energy values shown in Table 1 seem to be quite reasonable.

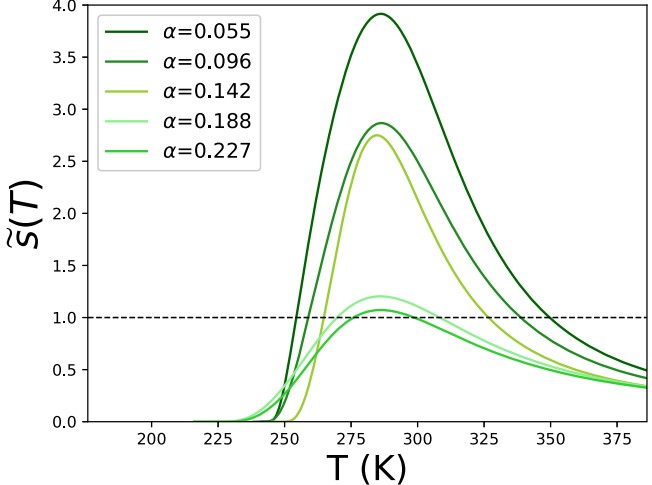

**Fig. 4 Stability parameter of Zimm–Bragg model with fitted parameters inserted.** Temperature dependence of renormalized stability parameter $\widetilde{s}$ from Eq. (3); color code of $\alpha = \frac{h_{ps} - h}{h}$ and other values as in Fig. 3d and Table 1d.

As any other approach, our model also has some natural limitations. For instance, volume changes have not been taken into account in our spin-based approach. Yet, Dias and Chan, in their simulation study using the TIP4P model of water have modeled the hydration effects on a pair of methanes[43] and came to the conclusion that the volume change can be either monotonic or nonmonotonic and can thus serve as ramifications for activation properties of protein folding. The comparison of our approach with molecular dynamics all-atom simulations may reveal, to which extent we mimic the pressure effects. Such a study is out of the scopes of the current paper and may be performed in the future.

Besides the applicative results we have reported, our results elucidate an important qualitative outcome: the delicate balance between inter- and intramolecular H-bonds determines the very existence of cold denaturation on the phase diagram of protein folding[26,27]. We show that cold denaturation is only possible when the water–polypeptide H-bonding is stronger than the intramolecular H-bonding ($\alpha > 0$). At the same time, if the water–polypeptide energy is too high, the system is denatured at any temperature. The absence of the cold denaturation discussed also in other theoretical models where the folding phase diagram is elliptic for all proteins[44], is here a consequence of the balance between inter- and intramolecular H-bond energies. Moreover, some proteins remain denatured, unless stabilized by osmotic stress or as a result of post-translational modifications. Nature may have used fine-tuning of protein sequences to provide the ability to be cold denatured upon necessity.

Last, we introduced water into the Zimm–Bragg picture at the level of Hamiltonian, thus reinforcing the foundations of a classical model. The explicitly formulated assumptions we have made enable controlling the level of approximation and the physical meaning of parameters. Armed with the formula for the order parameter, which in our original formulation depends on the water–polypeptide interaction constants, we provide, to the best of our knowledge, a new method to treat, process, and analyze the experimental data on protein folding.

## Methods
### Zimm–Bragg Hamiltonian with explicit water.
The Hamiltonian of our approach reads

$$H_{\text{total}}(\{\gamma_i, \mu_i^{(j)}\}) = H_{\text{Zimm–Bragg}}(\{\gamma_i\}) + H_{\text{water}}(\{\gamma_i, \mu_i^{(j)}\}), \tag{5}$$

where $\gamma_k = 1, 2, \ldots Q$ are spin variables, describing the conformations of each of $k = 1, 2, \ldots N$ peptide units, spin value $\gamma = 1$ corresponds to the ordered (helical) conformation, other $Q - 1$ values describe disordered (coil) conformations. $\mu_k^{(l)} = 1, 2, \ldots q$ spins describe water orientations around each peptide unit, two spins ($l = 1, 2$) per each broken H-bond reflect two binding sites for water; spin value $\mu = 1$ corresponds to water orientation, allowing for water-peptide H-bond, other $q - 1$ values are for disordered orientations. Zimm–Bragg Hamiltonian[33] can be written as

$$H_{\text{Zimm–Bragg}}(\{\gamma_i\}) = -U \sum_{i=1}^{N} \delta_i^{(2)}, \tag{6}$$

alone resulting in the partition function

$$Z_{\text{Zimm–Bragg}}(W, Q) = \sum_{\{\gamma_i\}} e^{-\beta H_{\text{Zimm–Bragg}}(\{\gamma_i\})} =$$
$$\sum_{\{\gamma_i=1\}}^{Q} \prod_{i=1}^{N} \left[1 + V\delta_i^{(2)}\right], \tag{7}$$

where $\delta_i^{(2)} = \delta(\gamma_i, 1)\delta(\gamma_{i+1}, 1)$, $V = W - 1$, $W = e^J$, $J = U/T$, $U$ is H-bond formation energy, $T$ is temperature and $N$ is the number of peptide units, considered large throughout the paper. We set $k_B = 1$ while deriving the formulas and recover its value once we start processing the experimental data. In the derivation of Eq. (7) we have used the Mayer's trick as in the cluster expansion[45]. Model with the Hamiltonian (6) can be treated with the transfer-matrix approach[33] and after the $\frac{\Lambda}{Q} \to \lambda$; $\frac{W-1}{Q} \to s$; $\frac{1}{Q} \to \sigma$ transformations leads to the original characteristic equation of the Zimm–Bragg model (see Supplementary Methods for details).

Second term of Eq. (5), describing the water–polypeptide interactions, reads

$$H_{\text{water}}(\{\gamma_i, \mu_i^j\}) = -E \sum_{i=1}^{N} (1 - \delta_i^{(2)})\left(\delta(\mu_i^{(1)}, 1) + \delta(\mu_i^{(2)}, 1)\right), \tag{8}$$

where $E(> 0)$ is the energy of a water–polypeptide H-bonding. It takes into account the fact that for intermolecular H-bonds with water to be formed, (i) the intramolecular H-bond should be broken first ($\delta_i^{(2)} = 0$), and (ii) the water molecules have to be in proper orientations around N−H and C=O groups ($\mu_i^{(j)} = 1$). A general case of Hamiltonian Eq. (8) differs from the past studies[21,26,27] in the range of H-bonds (considered nearest-neighbor here) which does not affect the summation over solvent degrees of freedom. Following past studies[21,26,27], the partition function of the model with Hamiltonian (5) reads

$$Z_{\text{total}} = \sum_{\{\gamma_i\}} \sum_{\{\mu_i^{(j)}\}} e^{-\beta H_{\text{total}}(\{\gamma_i, \mu_i^{(j)}\})} =$$
$$\sum_{\{\gamma_i\}} e^{-\beta H_{\text{Zimm–Bragg}}(\{\gamma_i\})} \sum_{\{\mu_i^j\}} e^{-\beta H_{\text{water}}(\{\gamma_i, \mu_i^{(j)}\})} =$$
$$\sum_{\{\gamma_i\}} \prod_{i=1}^{N} \left[1 + V\delta_i^{(2)}\right] \sum_{\{\mu_i^j\}} e^{-\beta H_{\text{water}}(\{\gamma_i, \mu_i^{(j)}\})}, \tag{9}$$

an expression, that contains the product of the unperturbed partition function Eq. (7) and a solvent-related term.

**From explicit water to implicit, effective model.** Although the total Hamiltonian (5) and the corresponding partition function (9) look complex, the independence of solvent degrees of freedom $\mu_i^{(j)}$ allows them to be analytically and explicitly summed out without invoking any approximation. Summation over solvent degrees of freedom and reordering of terms[21,26,27] results in

$$Z_{\text{total}} = Z(W, Q, K, q) = (K + q - 1)^{2N} \sum_{\{\gamma_i\}} \prod_{i=1}^{N} \left[1 + \widetilde{V}\delta_i^{(2)}\right], \tag{10}$$

where $K = e^{E/T}$, $\widetilde{V} = \widetilde{W} - 1$ and

$$\widetilde{W}(W, K, q) = \frac{q^2 W}{(q + K - 1)^2}. \tag{11}$$

Comparison of Eq. (10) with the partition function (7) of the in vacuo Zimm–Bragg model enables writing

$$Z(W, Q, K, q) \simeq Z_{\text{Zimm–Bragg}}(\widetilde{W}, Q). \tag{12}$$

The last result implies that we can effectively introduce water into the in vacuo Zimm–Bragg model by $W \to \widetilde{W}(W, K, q)$ substitution. The reason for the possibility of neglecting the pre-factor of the partition function lies in the definition of order parameter: both Eq. (10) and Eq. (12) give identical results when inserted into Eq. (16).

The result formulated in Eq. (12) enables replacing the explicit description of the water–polypeptide interactions with the implicit one by introducing effective, temperature-dependent energy

$$\widetilde{U}(T) = T\ln\widetilde{W}(T) = T\ln\left[\frac{q^2 e^{U/T}}{(q + e^{E/T} - 1)^2}\right]. \tag{13}$$

into the Hamiltonian of the in vacuo model, Eq. (6). To the best of our knowledge, this is the first implicit description of water–polypeptide interactions derived from the explicit model.

But what consequences can the $W \to \widetilde{W}(W, K, q)$ mapping have in terms of Zimm–Bragg parameters? Because of the

$$s \to \widetilde{s} = \frac{\widetilde{W} - 1}{Q}, \quad \sigma = 1/Q \quad (14)$$

relations between the two sets of model parameters (see Supplementary Methods and one of our past studies[33] for more details), the renormalization of $W$ affects only the stability parameter $s$, leaving the cooperativity parameter $\sigma$ of the Zimm–Bragg model intact.

**Theoretical and experimental grounds for temperature shift in Eq. (3).** Studies of relaxation phenomena in glass-forming liquids by default account for the shift in temperature by some value, corresponding to the glass formation temperature, $t_g$. In particular, $t - t_g$ temperature shift appears in hydrated proteins because of the presence of partially glassy states giving rise to non-Arrhenius relaxation times $\ln \tau \sim \frac{t_0}{t - t_0}$[46–48] ($t_0$ can be considered as having the meaning of glass transition temperature $t_g$ in supercooled liquid).

A statistical mechanics approach was suggested by Adam and Gibbs as early as in 1965 to describe the sudden increase of viscosity and the slowing down of the collective modes in supercooled liquids as $t \to t_0$[49]. The key idea of Adam-Gibbs theory was to consider the supercooled liquid as a set of clusters (cooperatively rearranging regions) of different sizes that change with temperature, giving rise to the $t - t_0$ shift in relaxation time. It was reported, that for the number of glass-forming liquids, there are two interesting relations between the second-order transition temperature ($t_2$) and $t_0$, namely, $t_2/t_0 \simeq 1.3$ and $t_2 - t_0 \simeq 50\,°\mathrm{C}$. Interestingly, if instead of $t_2$ the helix-coil transition temperature is considered, similar values for the relations result (see Fig. 3).

The temperature shift factor is present in many theories describing properties of water. Thus, Truskett and Dill had to include the Adam-Gibbs temperature shift into their simple analytical model of water to achieve the agreement with experimental data on the temperature dependence of self-diffusion coefficient[50].

Later, Schiro and Weik have summarized recent in vitro and in silico experimental results regarding the role of hydration water in the onset of protein structural dynamics[51], and have reported the presence of super-Arrhenius relaxation region above the protein dynamic transition temperature.

Recently, Mallamace et al. have used the Adam-Gibbs theory in their NMR measurements of protein folding–unfolding in water[52,53] in order to rationalize the complicated pressure-temperature diagrams in these glass-forming systems.

Motivated by the considerations above, and taking into account the $\widetilde{s}(t) \sim \tau^{-1}$ relationship between the unimolecular rate of folding in water and the relaxation time[54], we introduce the $t - t_0$ temperature shift into Eq. (14) through Eq. (3) and finally, to Eq. (4) used to fit experimental data on hydrated polypeptides. For the sake of Occam's razor, we have also checked the fit without temperature shift and found it never converges for all data sets checked.

**Entropic cost of H-bonding with water.** According to the definition of H-bonds, summarized by International Union of Pure and Applied Chemistry (IUPAC)[55,56], besides relevant distance constraints, the angle between the hydrogen bond donor X−H and acceptor Y has to be close to the straight line (180°). However, if, in the case of water, the angle O−H ⋯ O is above 110°, then the angle HOO (used in the study[39], which we follow for the definition of $q$) cannot be larger than 35 or 40°. This, would imply that the value of $q$ should be between $5^2 = 25$, from $360/(35*2) = 5$, and $4^2 = 16$, from $360/(40*2) > 4$. Here we have the hydrogen bond between water and N−H or C=O groups of the peptide, which is bulkier than water. Taking into account all the above, our choice of the parameter value is

$$q = 4^2 = 16. \quad (15)$$

**Final expressions, used in fit.** Final expressions, used in fit are summarized here for reference purposes (see Supplementary Methods for the detailed derivation). Order parameter (helicity degree):

$$\theta_{\text{Zimm–Bragg}}(\widetilde{s}, \sigma) =$$

$$\frac{\widetilde{s} + \sigma}{1 + \widetilde{s} + \sqrt{(1 - \widetilde{s})^2 + 4\sigma\widetilde{s}}} \left( 1 + \frac{2\sigma - 1 + \widetilde{s}}{\sqrt{(1 - \widetilde{s})^2 + 4\sigma\widetilde{s}}} \right), \quad (16)$$

where

$$\widetilde{s}(t, t_0, h, h_{ps}, Q, q = 16.0) =$$

$$\frac{1}{Q} \left[ \left( e^{-h/(t - t_0)} + \frac{e^{(h_{ps} - h)/(t - t_0)} - e^{-h/(t - t_0)}}{q} \right)^{-2} - 1 \right], \quad (17)$$

and $\sigma = 1/Q$.

## Data availability

Relevant data are available from the corresponding author upon reasonable request.

## Code availability

Available upon request to Artem Badasyan.

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

## Acknowledgements
Miss Knarik Yeritsyan has collaborated during the initial stages of the research presented above. Useful discussions with Dr. Franc Avbelj, Dr. Giancarlo Franzese, Dr. Luigi Giacomazzi, Dr. Yevgeni Mamasakhlisov, Dr. Vladimir Morozov, and Dr. Rudolf Pod-gornik are thankfully acknowledged. Sh.T. acknowledges partial financial support from RA MES State Committee of Science through the research project 16YR-1F046. A.B. acknowledges the partial financial support from Erasmus+ project 2018-1-SI01-KA107-046966; A.B. and J.G. acknowledge the partial financial support from the Slovenian Research Agency through project J1-1705; A.B. and M.V. acknowledge the partial financial support from the Slovenian Research Agency through program P2-0412.

## Author contributions
A.B. designed and developed the overall method and approach. A.B. and J.G. supervised the research. A.B. and S.T. wrote the code and analyzed the data. A.B., J.G., and M.V. wrote the paper. All authors read and commented on the paper.

## Competing interests
The authors declare no competing interests.
