## [Peer Review File · Communications Chemistry]

Reviewers' comments:

Reviewer #1 (Remarks to the Author):

The manuscript by Badasyan et al. describes a generalization of the Zimm-Bragg model including water-polypeptide interactions. The model is benchmarked on 4 test cases and shows good agreement with experimental data.

This contribution has the merit to shed a light at an early, yet fundamental, minimalistic model of protein folding. The proposed framework allows the authors to capture interaction with water from experimental data, which is a useful feature. Overall, I like the approach that revisits a simple but important model. However, I think the manuscript could be improved.

Major comments:

- The presentation/introduction should be improved. In particular, the Zimm-Bragg model is not properly introduced for a reader not familiar with it (e.g. the parameters s and Δ are used and discussed but not formally introduced). From my perspective, it would be essential to include a figure illustrating the original and proposed model.
- The legend and annotation of figures is not clear or detailed. I would suggest the author to improve the caption and legend for clarity and better interpretation of the data. For instance, it was not obvious to me how to read "... the statistical weight of growth of ordered conformation in presence of water, as can be seen in Figure 2a" (page 10).
- The observations about the estimates of h-bond energies are interesting and would deserve to be more discussed/investigated. One limitation of this manuscript is that the number of experimental systems on which the model is tested is limited. Larger scale studies might not be required for this manuscript, but I think a better/deeper illustration of the (novel?) insights offered by the model would be profitable for the reader.

Minor comments:

- On page 4, "How to extract the constants of the interaction with water from the experimental data?" I think it would be nice to justify in the text why this is important.
- Add a citation for the "Mayer trick" (page 5)
- Add citation for "theoretical studies of hydrated proteins report the need for introducing the temperature shift to reflect the super-Arrhenius relaxation observed in the experiment" (page 10)
- Wording could be improved in several places (e.g. "Model with the Hamiltonian (2), treated with the transfer-matrix approach" on p. 5)
- Missing reference in "See Methods" on page 10.

Reviewer #2 (Remarks to the Author):

This theoretical development is very interesting and useful for connecting experimental observation of helix-coil transitions to theoretical/basic energetic parameters. The formulation appears to be well executed. This work should definitely be published. However, there is room for improvement with regard to presentation of the authors' results as well as placement of their work in the broader scientific context. Accordingly, this manuscript should be revised to address the following:

1. On page 5, the symbols (upper case Greek letter "Lambda") and (lower case Greek letter beta) are undefined. Is beta equal to $1/T$? If so, one doesn't need the symbol J (one can just use U).
2. Page 5: what's the physical meaning of "two spins ($l = 1, 2$) per broken H-bond"?

3. Page 5: Could the authors provide a brief sentence or two to explain the quantity Q , what's expected value(s), and what's the corresponding physical basis? How realistic is this kind of abstract spin-based water models? Please mention briefly any previous comparison with experiment and/or explicit-water MD simulations.
4. Page 7, line after Eq.8: "implays" should be "implies" (typo)?
5. The proposed theory-experiment fit requires four fitting parameters. Would that be too many, as the quality of the fit is bound to get better with more fitting parameters? The question is what physics is being captured. In Eq.4 on page 11, authors should briefly comment on whether (i) t_0 is obtained from independent experimental measurement, and if so, what experiment, and (ii) why h , h_{ps} , and Q are not constant for different helix-coil systems.
6. The authors commented at the bottom of page 11 that the fit in Fig.3 of Ref.28 was obviously very poor. It would be useful to reproduce that fit here (e.g., as an inset to Fig.3a of the present manuscript) to make it easier for readers to grasp the difference. Authors should also comment on how many fitting parameters were used in this prior "poor" fit.
7. Pages 20-24: Somehow the Methods section comes between the references. Please check.
8. Fig.1: What is the meaning of the red and blue color ("native" and "denatured")? They should be defined clearly, with a color scale if necessary. Moreover, what exactly is the definition of "native" and "denatured" (inset) in this context? Do "native" and "denatured" refer to a single unit or a chain? If a chain, what's the chain length (if the theory is independent of chain length in some long chain length limit, it should be stated). Is "native" and "denatured" related to theta (fractional helicity)?
9. Fig.3: It would be helpful to state the source of experimental data in the caption, and specify which data points are what.
10. A very brief discussion should be included to address the relationship of the present approach with explicit-water simulation (this is very do-able these days for short helices), as well as the relationship between the authors' spin-derived temperature-dependent effective interactions and the temperature-dependent hydrophobic effects obtained from transfer experiments and explicit water molecular dynamics simulations [Dill et al., *Biochemistry* 28:5349 (1989); Dias and Chan, *J Phys Chem B* 118:7488 (2014)]. Include these two references in the added discussion.

Reviewer #3 (Remarks to the Author):

The authors use a new method to analyze experimental data for protein heat and cold denaturation. The method is based on a Hamiltonian formulation of the explicit-water Zimm-Bragg model, with a term for the hydration water, that some of them have shown to be equivalent to the implicit-water Zimm-Bragg model with effective energy constants. Here they show that the order parameter of the implicit-water model fits well several experimental sets of data, covering a variety of cases of interest. Specifically, they consider four cases (poly-L-alanine homopolypeptide, natural recombinant human Apo A-1 protein, and lysozyme, intrinsically disordered yeast frataxin (Yfh1), and α 3C de novo designed three-helix-bundle protein), with the last two displaying heat and cold denaturation. The latter is particularly interesting because it has, or has not, cold denaturation depending on a denaturant concentration.

The authors show that in all cases their fitting methods gives very good agreement with the experimental data, with fitting parameters that are within a reasonable range of values. Among these parameters, quite interesting are the energies estimates for the average polypeptide-water,

or inter-molecular, hydrogen bonds (HBs) and the average energy of the water-water (w-w) and polypeptide-polypeptide (p-p), or intra-molecular, HBs. The estimates of these energies allows the authors to conclude that the folding and cold denaturation are possible only when the balance among these energies is within the appropriate range.

This conclusion is, in my understanding, original. Indeed, even if other works, including some of those cited by the authors, have shown the relevance of HBs balance to get cold denaturation in theoretical models, these previous works focused on the balance between different kind of HBs, e.g., w-w vs. p-p, but not intra vs. inter-molecular HBs.

I believe that this fitting method is quite interesting. It helps to extract relevant information from the experimental data and to discuss fundamental mechanisms regulating the folding-unfolding phenomena. The fitting method, although derived from a model studied in previous publications, is novel, with results that are original and of interest to (experimental and theoretical) researchers in the field of protein folding and to the wider community of researchers studying multidisciplinary applications of statistical physics.

The work is convincing, with a statistical analysis that is scientifically sound. The method could have an important impact in the field. The level of detail provided is enough for other researchers to reproduce the work, however there is a point that should be discussed further.

- On pages 22-23, they discuss the value for the water parameter q , related to the number of H-bonded or non-bonded states. They choose $q=3^2=9$ and refers to the Footnote F4 of Ref. 46, which says: "The X-H□□□...Y hydrogen bond angle tends toward 180° and should preferably be above 110° ". However, if, in the case of water, the angle O-H□□□...O is above 110° , then the angle HOO (used in Ref.33, which the authors follow for the q definition) cannot be larger than 35 or 40° . This, would imply that the value of q should be between $5^2=25$, from $360/(35*2)\sim 5$, and $4^2=16$, from $360/(40*2)>4$. Their choice $q=3^2$ implies, instead a O-H□□□...O angle as small as 45° , in contradiction with the Footnote F4 of Ref. 46 (and all the rest of references I know).

The selection $q=6^2=36$, consistent with Ref.33, would coincide with the standard definition of a $\pm 30^\circ$ deviation from the straight HB, as used by Luzar and Chandler in <https://doi.org/10.1103/PhysRevLett.76.928>, derived from the experiments made by Teixeira, Bellisent-Funel, and Chen in <http://iopscience.iop.org/0953-8984/2/S/011>, and confirmed many times (see, e.g, <https://doi.org/10.1039/C9CP04795F> for a recent quantum calculation). A review of many works discussing the angle cut-off for the HB can be found in the section "Water hydrogen bond direction" of the web page http://www1.lsbu.ac.uk/water/water_hydrogen_bonding.html#length cured by Martin Chaplin.

I believe that the mismatch between the choice $q=3^2$ and $q=6^2$ could be due to a misunderstanding of the angle definitions, where the O-H□□□...O angle and the HOO angle are confused (e.g., see the wrong use of the "O-H□□□...O angle" terminology used in <https://doi.org/10.1103/PhysRevLett.76.928> when referring, instead, to the HOO angle).

Hence, the authors should discuss in more details how their results depend on the choice $q=3^2$, even if the HB definition is beyond the scope of their study.

Did they made calculations using $q=4^2$, 5^2 , or 6^2 ?

They only write: "we have also tried a fit without fixed value of q , and found only slight deviations (about 10%) from $q = 9$ estimate".

This observation should be elaborated in more details, explaining how these alternative fits were done, e.g., which parameters were free and which were fixed and how. A possible outcome could

be that the physical interpretation of the parameter q is different from what they present here.

In general, the presentation is clear, although it could be improved in a few points. For example,

- The adverb "perfectly", in the sentence "Our model perfectly fits to the Circular Dichroism experimental data..." of the abstract, could be replaced with "Our model fits very well to ..."

- In the introduction they mention that previous fitting methods "result in non-matching sets of fitted parameter values for Circular Dichroism (CD) and calorimetric data of the same protein". It is not clear to me is their method, instead, is able to match both CD and calorimetric data of the same protein. Can they comment in the manuscript about this point and, if possible, show it with a plot?

- Although it is true that the anomalies of water are more evident below 4C, the statement that "in the physiologically relevant range and close to normal conditions, nothing decisive happens with water itself" should be emended, because it is not correct. Water at ambient condition has the isothermal compressibility and the isobaric heat capacity that increase for decreasing temperature, at variance with normal liquids. These anomalous behaviours of the response functions could be essential to understand the relevance of water in biological systems, because they are related to the anomalous behaviour of water's free energy (see, for example, https://doi.org/10.1007/978-3-540-78765-5_1 for a short review).

- The statement "the water-polypeptide interactions mostly span up to the second water layer" should be supported at least with a reference, and it should be mentioned that it is known that the effect of biological interfaces can span much more than two layers of water (see, for example, <https://dx.doi.org/10.1021/acsnano.0c02984> for models and <http://dx.doi.org/10.1103/PhysRevLett.106.158102> for experiments). Hence, the statement should be considered as a (reasonable) approximation.

- There are mismatches of notations and formulas between the main text and the SI, e.g., Eq.(2) in the main text and Eq.(8) in SI, or the definition of W . Furthermore, often (but not always) the Boltzmann constant is (unnecessarily) set to 1 without mentioning it. Also, they use the notation \log instead of the more appropriate \ln , for the Neperian logarithm. Furthermore, it would be helpful to use different symbols for the "=" in equations and "=" for definitions. All these notation issues make difficult to follow the discussion.

- When they introduce the reduced temperature τ , it would be better to use parenthesis to avoid misunderstandings. Also, they should clarify the advantage of using the factor $\log Q$ in its definition.

- When describing the explicit-water model on page 5 of the main text, they should clarify better why each broken HB has only two spins described the interaction with water. If I understand correctly, it is because the authors consider the (one-dimensional) polypeptide chain embedded into two dimensional water. In any case, apart from this, i) they should clarify the dimensionality of the model at the beginning of its definition, ii) they should mention somewhere if they expect any dependence of their results on the dimensionality of the model and why.

- Is their model including any effect of the protein interface on the w-w interaction? Up to Eq.(12) I was convinced that there was no w-w interaction. However, in Eq.(12) the energy $U_{\{ss\}}$ refers to a w-w interaction. Then, I must have missed the description of the effect. In any case, they should make a comment about it, because in other models this effect is shown to be very relevant (see, e.g., their ref. 22, 23 or <http://dx.doi.org/10.1103/PhysRevLett.103.037803>). If in their model there is no w-w interaction, then I wonder if the introduction of $U_{\{ss\}}$ and $U_{\{pp\}}$, just before Eq.(12), is necessary.

- Eq. (9) shows that the renormalized p-p energy coincides with the original U (i.e., the w-p interaction has no effect) in the limit of large q, i.e., if the entropy of water is very large. Can the authors comment on the meaning, within their model, of this limit?
- In the sentence at the beginning of page 9 they use the expression "critical behavior" (and later "transition"), but, in my understanding, there is no phase transition in the model. They should replace this expression with something more precise, because critical behavior in this context would generate misunderstanding.
- On page 9 "[...] the energy dependence of s_{\square} at different temperatures, which becomes non-bijective for $a < 0$ ", should be "... $a > 0$ ".
- On page 11, before Eq.(14) is the correct reference to Eq.(18), instead of (16)?
- On page 13, is the sentence " between the two critical points of water" referring to the liquid-gas and the (hypothesised) liquid-liquid critical point? Please, clarify.
- In the first sentence on pag. 7 of the SI, "and" should be replaced with "," .

In conclusion, this manuscript presents a work that is interesting and worth to be published. However, the authors should consider the points indicated in this report and they should resubmit a revised version, before the manuscript can be accepted.

Please find below our point-by-point answer to Reviewer's comments.

Reviewers' comments:

Reviewer #1 (Remarks to the Author):

The manuscript by Badasyan et al. describes a generalization of the Zimm-Bragg model including water-polypeptide interactions. The model is benchmarked on 4 test cases and shows good agreement with experimental data.

This contribution has the merit to shed a light at an early, yet fundamental, minimalistic model of protein folding. The proposed framework allows the authors to capture interaction with water from experimental data, which is a useful feature. Overall, I like the approach that revisits a simple but important model. However, I think the manuscript could be improved.

AUTHORS: Thank you for your kind words. We will do our best to improve the text and answer each of your comments. Parts of text and formulas that has been altered are coloured green. One general reply to your comments: the format of Nature group of journals requires a very structured text of paper, with Methods section immediately after the main text, and with separate list of references, and a separate Supporting Info file, again with its own list of references. Therefore, we quite often redirect to those sections and explain details there.

Reviewer #1: Major comments:

â€¢ The presentation/introduction should be improved. In particular, the Zimm-Bragg model is not properly introduced for a reader not familiar with it (e.g. the parameters s and Δ are used and discussed but not formally introduced). From my perspective, it would be essential to include a figure illustrating the original and proposed model.

AUTHORS: On p.4, last paragraph, we say:

"The Zimm-Bragg model formulated in the 1950s, is so successful in describing conformational transitions in biopolymers that it is still widely used to treat experimental data. In addition to its strengths, the original model formulation (see Supplementary Info) also contains essential weaknesses, mostly due to the absence of the microscopic Hamiltonian. Indeed, it is not clear how the parameters s and Δ should be amended to describe solvent effects..."

So that the reader is directed to consult the ZB model in the Supplementary Info (SI). If done so, the meaning of the parameters s and Δ becomes clear. SI is almost fully intended to different formulations and aspects related to Zimm-Bragg model, including the definition of parameters. We show in SI, that both definitions have identical eigenvalues upon the mapping we suggest (see SI Eq.14), so that there is no difference between the original and proposed model (please compare SI Eq.2 with SI Eq.15).

Reviewer #1 $\hat{\phi}$ The legend and annotation of figures is not clear or detailed. I would suggest the author to improve the caption and legend for clarity and better interpretation of the data. For instance, it was not obvious to me how to read $\hat{\phi}$ the statistical weight of growth of ordered conformation in presence of water, as can be seen in Figure 2a $\hat{\phi}$ (page 10).

AUTHORS: We have corrected the legends and annotations of figures, and moved explanatory text from the captions to those parts in the body of manuscript, where the figures are first discussed. The text on page 10, stating $\hat{\phi}$ the statistical weight of growth of ordered conformation in presence of water... $\hat{\phi}$ is modified to read now:
"...the Zimm-Bragg stability parameter in water..."

Reviewer #1 $\hat{\phi}$ The observations about the estimates of h-bond energies are interesting and would deserve to be more discussed/investigated. One limitation of this manuscript is that the number of experimental systems on which the model is tested is limited. Larger scale studies might not be required for this manuscript, but I think a better/deeper illustration of the (novel?) insights offered by the model would be profitable for the reader.

AUTHORS: We completely agree with the Reviewer's observations. However, we work on the second MS where our own data of protein unfolding will be used and there we will describe the calculated energies in more details. Guided by the Reviewer's comment we have amended the Introduction as well as the Conclusion sections.

Reviewer #1 Minor comments:

$\hat{\phi}$ On page 4, $\hat{\phi}$ How to extract the constants of the interaction with water from the experimental data? $\hat{\phi}$ I think it would be nice to justify in the text why this is important.

AUTHORS: We agree with the Reviewer so we have amended the Introduction by including several paragraphs explaining the relevance of interaction constants.

Reviewer #1 $\hat{\phi}$ Add a citation for the $\hat{\phi}$ Mayer trick $\hat{\phi}$ (page 5)

AUTHORS: We have rewritten this part of text and added a citation:

"In the derivation of Eq. (3) we have used the Mayer's trick as in the cluster expansion 27."

27. Mayer, E.M. & Mayer, G.M. Statistical Mechanics (J. Wiley & Sons, New York, 1940).

Reviewer #1 $\hat{\phi}$ Add citation for $\hat{\phi}$ theoretical studies of hydrated proteins report the need for introducing the temperature shift to reflect the super-Arrhenius relaxation observed in the experiment $\hat{\phi}$ (page 10)

AUTHORS: On page 10, above Eq.13, a paragraph you have partially cited, says:

"Recently, theoretical studies of hydrated proteins report the need for introducing the tem-

perature shift to reflect the super-Arrhenius relaxation observed in the experiment. Thus, $t \hat{=} t_0$ temperature shift appears in hydrated proteins, because of the presence of partially glassy states; t_0 is a fitting parameter, that represents the glass transition temperature in supercooled liquid (see Methods)."

In Methods (please consult pages 21-22), there is a chapter devoted to the clarification of the statement and its discussion and overview, with references 37-45 related to exactly this topic.

Reviewer #1

Wording could be improved in several places (e.g. ``Model with the Hamiltonian (2), treated with the transfer-matrix approach'' on p. 5
AUTHORS: We have changed the sentence to sound:

"Model with the Hamiltonian (2) can be treated with the transfer-matrix approach (see Ref....) and after the..."

We have also done our best improving wording throughout the manuscript (see changes in green in the marked-up file).

Reviewer #1 Missing reference in ``See Methods'' on page 10.

AUTHORS: Thank you for the comment. The reference appears before the label, we have deleted it. Methods section spans 3 pages and the corresponding subsection can be easily found.

CONCLUDING REMARK FROM THE AUTHORS: We thank the Reviewer for useful and constructive comments which we tried to follow as much as possible, giving explanation whenever disagreed.

Reviewer #2 (Remarks to the Author):

This theoretical development is very interesting and useful for connecting experimental observation of helix-coil transitions to theoretical/basic energetic parameters. The formulation appears to be well executed. This work should definitely be published. However, there is room for improvement with regard to presentation of the authors' results as well as placement of their work in the broader scientific context.

AUTHORS: Thank you for your kind words. We will do our best to improve the text and answer each of your comments. Parts of text and formulas that has been altered are coloured green. One general reply to your comments: the format of Nature group of journals requires a very structured text of paper, with Methods section immediately after the main text, and with separate list of references, and a separate Supporting Info file, again with its own list of references. Therefore, we quite often redirect to those sections and explain details there.

Reviewer #2 Accordingly, this manuscript should be revised to address the following:

1. On page 5, the symbols (upper case Greek letter Λ) and (lower case Greek letter β) are undefined. Is β equal to $1/T$? If so, one doesn't need the symbol J (one can just use U).

AUTHORS: The text at the end of page 5 and continued on page 6 says:

"Model with the Hamiltonian (2), treated with the transfer-matrix approach (see Ref.27), after the ... transformations leads to the original characteristic equation of the Zimm-Bragg model (see Supplementary Info for details)."

In Supplementary Info (SI), pages 2-7 we give detailed definitions of all the parameters, including the symbols (upper case Greek letter λ) (please consult the text above SI Eq.11) and (lower case Greek letter β) (please consult the text below SI Eq.8). The use of symbol J is relevant in view of Statistical Mechanics, where it is often called as "coupling". Thanks to your comment we have noticed a typo in SI Eq.8, 10 and 11, as well as in the text after Eq.3 of main text. It is a typo in the presentation, not in the actual calculation and didn't affect final results.

Reviewer #2 2. Page 5: what's the physical meaning of two spins ($l = 1, 2$) per broken H-bond?

AUTHORS: To address your request we have modified the text to: "two spins ($l = 1, 2$) per each broken H-bond to reflect two binding sites for water".

Reviewer #2 3. Page 5: Could the authors provide a brief sentence or two to explain the quantity Q , what's expected value(s), and what's the corresponding physical basis? How realistic is this kind of abstract spin-based water models? Please mention briefly any previous comparison with experiment and/or explicit-water MD simulations.

AUTHORS: In SI, page 6, above SI Eq.8 there is some text, saying: "Assume Q (≥ 2) possible values for the spin \hat{I}^i describing the state of the i -th repeated unit; $\hat{I}^i = 1$ value corresponding to values of the torsional angles $\hat{\tau}^i$ and $\hat{\tau}^i$ from the helical region of the Ramachandran map, while the other $Q - 1$ identical values correspond to torsional angles from the coil region. In brief, Q has the meaning of the phase space of our problem."

We believe it answers your question.

The word "volume" has been added in the last sentence to sound "In brief, Q has the meaning of the phase space volume of our problem."

Comparison with experiment is only possible after at least water is introduced and is for the first time done in this paper. SI Eq.14 shows that it is related to the parameter σ of Zimm-Bragg model through $\sigma = 1/Q$ mapping.

Reviewer #2 4. Page 7, line after Eq.8: "implays" should be "implies" (typo)?

AUTHORS: Typo corrected, thank you.

Reviewer #2 5a. The proposed theory-experiment fit requires four fitting

parameters. Would that be too many, as the quality of the fit is bound to get better with more fitting parameters? The question is what physics is being captured.

AUTHORS:

Regarding the number of fitting parameters. A minimal model we develop should at least include waters and a protein. Everything is defined by the tiny balance between energies of interactions and corresponding entropies. Immediately we come to 4 parameters. Besides, a temperature of glass formation should be taken into account, thus 5 parameters. Luckily, water is well-studied and conceptually following other studies we could estimate the entropy of H-bond formation as $\Delta q = 16.0$. Thus, fit reduced to 4 parameters. For instance, in J. AM. CHEM. SOC. 2009, 131, 2306-2312 by Ghosh and Dill, five fitting parameters are used and the approach still does not allow to describe cold denaturation. In Go, M., Go, N. & Sheraga, H.A. J. Chem. Phys. 54, 4489-4503 (1971), a large (>5) number of molecular quantities are used to estimate temperature dependencies of Zimm-Bragg σ and σ variables, still leading to a very poor agreement with the experiment (see Figure 3 of their paper). Thus increasing the number of parameters does not always bring to the improvement of final result, by the way. In J. Phys. Chem. B 2012, 116, 8095-8104, by Matysiak et al considers at least 7 parameters to describe water explicitly. We suggest a minimal model, that, on hand, allows to account for water explicitly, reduce it to the implicit description, and provide the fit with 4 parameters, that reproduces the experimental data. Usual 2-state fit with Hawley-like formula is a 2 parameter fit (in fact, 3, if take into account the ill-defined transition temperature if transition curve doesn't reach 100% height), but there it says nothing about the interaction parameters with water. Thus our recipe better captures the physics and is tractable from the Hamiltonian to fitted formulas.

Reviewer #2 5b. In Eq.4 on page 11, authors should briefly comment on whether (i) t_0 is obtained from independent experimental measurement, and if so, what experiment, and (ii) why h , h_{ps} , and Q are not constant for different helix-coil systems.

AUTHORS: In response to your request, we have added a sentence below Eq. 14 on page 11:

"We do not exclude the possibility to determine some of these parameters from the independent measurements or simulations, but this question is out of the scopes of the presented study."

It is not a simple question and since we didn't make any studies allowing to make justified conclusions on the question, we do not feel eligible to discuss this point; it would be truly speculative otherwise. Regarding the different resulting parameters h , h_{ps} , and Q , proteins are different and have unique sequence, therefore the effective homopolymer model we use results in different parameters for each protein. Actually, we are not aware of a heteropolymer approach attempting to process experimental data. Such a discussion in the text of our paper would take a reader far away from the main focus of our paper. With this reasoning we didn't add it.

Reviewer #2 6. The authors commented at the bottom of page 11 that the fit in Fig.3 of Ref.28 was obviously very poor. It would be useful to reproduce

that fit here (e.g., as an inset to Fig.3a of the present manuscript) to make it easier for readers to grasp the difference. Authors should also comment on how many fitting parameters were used in this prior "poor" fit.

AUTHORS: Fig3a is very dense and the quality of reproduced picture is very low. To improve the visual comparison we have decided to repeat Figure 3a side by side with Figure 3 of Go, M., Go, N. & Sheraga, H.A. J. Chem. Phys. 54, 4489-4503 (1971) and show it in the Supporting Info section. The number of fitting parameters is large and includes a number of molecular characteristics reported in a series of papers, making it very hard to count. Once we have noticed it is larger than 5, we stopped following.

Reviewer #2 7. Pages 20-24: Somehow the Methods section comes between the references. Please check.

AUTHORS: Thank you for the comment. Such a structure is the result of the Nature group requirements we have to obey.

Reviewer #2 8a. Fig.1: What is the meaning of the red and blue colour ("native" and "denatured")? They should be defined clearly, with a colour scale if necessary.

AUTHORS:

In Figure 1 the red and blue colours are used for the illustrative purposes to show the texture of the surface in 3D. In the bottom projection and in the inset the colour code is used to show the regions of phase diagram, separated by ~ 1 line. In Figure 2, where the colour code is related to particular values of parameters, there is a scale present.

Reviewer #2 8b. Moreover, what exactly is the definition of "native" and "denatured" (inset) in this context?

AUTHORS: In captions to Figure 1 we clearly say: "... ~ 1 separates the native conformations from the denatured...".

Reviewer #2 8c. Do "native" and "denatured" refer to a single unit or a chain? If a chain, what's the chain length (if the theory is independent of chain length in some long chain length limit, it should be stated). Is "native" and "denatured" related to theta (fractional helicity)?

AUTHORS: Thank you for the question. Below Eq.3 on page 5 we have added "...N is the number of peptide units, considered large throughout the paper."

Zimm-Bragg model, when defined without the Hamiltonian, indeed opens up many questions. The presence of Hamiltonian resolves this point. On page 6 of SI, in the section "HAMILTONIAN FORMULATION OF ZIMM-BRAGG MODEL", above Eq. 8 we clearly define at the level of each repeat unit what does it mean helical, and what coil (microstate). The Hamiltonian Eq. 8 shows that each repeat unit contributes to system energy, since summation is from 1 to N . Partition function Eq. 9 builds up the statistical description using the transfer-matrix approach. Resulting Eq. 18 is the helicity degree as the order parameter, which changes in the range between 0 and 1. Whenever the helicity degree is larger than 0.5, we refer to such case as a helical

macrostate, and below 0.5 is the coil. Please pay attention, that in Zimm-Bragg model the temperature dependence of the model is described by parameter s (see also Eq.18, SI and Figure 1, SI). Therefore, although by its meaning s parameter can be considered as the probability to find the repeat unit in the helical conformation (provided that the previous repeat unit is also helical), its temperature dependence can be used to distinguish between the chain conformations. We have used $s=1$ as a border separating the two macroscopic conformations, helical and coil. One could ask, how these two are related to native and denatured. Leaving aside the question, what happens first, the helix-coil transition or protein denaturation, we notice that both phenomena take place in the same interval of external variables, temperature, in this case. Besides, UV-VIS mostly captures changes in the secondary structure. We have checked the text of the main file and the Supporting Info, and believe they contain the answers to all of above comments. In particular, on page 8 of SI, above Eq. 17, we clearly say: "The order parameter (helicity degree) can be defined as the average relative number of intramolecular H-bonds..."

Reviewer #2 9. Fig.3: It would be helpful to state the source of experimental data in the caption, and specify which data points are what.

AUTHORS: As requested, we have added the info on the sources of experimental data into the captions of Figure 3.

Reviewer #2 10. A very brief discussion should be included to address the relationship of the present approach with explicit-water simulation (this is very do-able these days for short helices), as well as the relationship between the authors' spin-derived temperature-dependent effective interactions and the temperature-dependent hydrophobic effects obtained from transfer experiments and explicit water molecular dynamics simulations [Dill et al., Biochemistry 28:5349 (1989); Dias and Chan, J Phys Chem B 118:7488 (2014)]. Include these two references in the added discussion.

AUTHORS: In response to your request, we have added both references and the following text to Conclusions:

On pp.15-16:

"

As any other approach, our model also has some natural limitations. For instance, volume changes have not been taken into account in our spin-based approach.

Yet, Dias and Chan, in their simulation study using the TIP4P model of water

have modelled the hydration effects on a pair of methanes

\cite{diaschan} and came to the conclusion that the volume change can be either monotonic or

non-monotonic and can thus serve as ramifications for activation properties of protein folding. The comparison of our approach with molecular dynamics all-atom simulations may reveal, to which extent we mimic the pressure effects. Such a study is out of the scopes of the current paper and may be performed in the future.

"

On p.3:

"

An interesting

paper by Dill et al, devoted to the investigation of the hydrophobic effect, have explained the cold denaturation as appearing due to the "weakening" of the interactions with the solvent 14. Their thermodynamic mean-field theory widely uses Flory-Huggins-like approach with the effective energy of "solvophobic" interaction, defined by the Hawley 13 formula and reaches the qualitative agreement with some calorimetric experiments.

"

CONCLUDING REMARK FROM THE AUTHORS: We thank the Reviewer for useful and constructive comments which we tried to follow as much as possible, giving explanation whenever disagreed.

Reviewer #3 (Remarks to the Author):

The authors use a new method to analyse experimental data for protein heat and cold denaturation. The method is based on a Hamiltonian formulation of the explicit-water Zimm-Bragg model, with a term for the hydration water, that some of them have shown to be equivalent to the implicit-water Zimm-Bragg model with effective energy constants. Here they show that the order parameter of the implicit-water model fits well several experimental sets of data, covering a variety of cases of interest. Specifically, they consider four cases (poly-L-alanine homopolypeptide, natural recombinant human Apo A-1 protein, and lysozyme, intrinsically disordered yeast frataxin (Yfh1), and ^{13}C de novo designed three-helix-bundle protein), with the last two displaying heat and cold denaturation. The latter is particularly interesting because it has, or has not, cold denaturation depending on a denaturant concentration.

AUTHORS: We thank Reviewer 3 for the opinion. The last comment holds for the case of $q=9$, but not for the $q=16$, which was suggested by the Reviewer (and accepted by us). We make proper changes and mention them below.

Reviewer #3

The authors show that in all cases their fitting methods gives very good agreement with the experimental data, with fitting parameters that are within a reasonable range of values. Among these parameters, quite interesting are the energies estimates for the average polypeptide-water, or inter-molecular, hydrogen bonds (HBs) and the average energy of the water-water (w-w) and polypeptide-polypeptide (p-p), or intra-molecular, HBs. The estimates of these energies allows the authors to conclude that the folding and cold denaturation are possible only when the balance among these energies is within the appropriate range.

AUTHORS: We appreciate the detailed reading of our results.

Reviewer #3

This conclusion is, in my understanding, original. Indeed, even if other works, including some of those cited by the authors, have shown the relevance of HBs balance to get cold denaturation in theoretical models, these previous works focused on the balance between different kind of HBs, e.g., w-w vs. p-p, but not intra vs. inter-molecular HBs.

AUTHORS: We also find it relevant, thanks.

Reviewer #3

I believe that this fitting method is quite interesting. It helps to extract relevant information from the experimental data and to discuss fundamental mechanisms regulating the folding-unfolding phenomena. The fitting method, although derived from a model studied in previous publications, is novel, with results that are original and of interest to (experimental and theoretical) researchers in the field of protein folding and to the wider community of researchers studying multidisciplinary applications of statistical physics.

The work is convincing, with a statistical analysis that is scientifically sound. The method could have an important impact in the field. The level of detail provided is enough for other researchers to reproduce the work, however there is a point that should be discussed further.

AUTHORS: We also believe the results can shift the paradigm in the field.

Reviewer #3

- On pages 22-23, they discuss the value for the water parameter q , related to the number of H-bonded or non-bonded states. They choose $q=3^2=9$ and refers to the Footnote F4 of Ref. 46, which says: "The $\angle \text{H}\dots\text{Y}$ hydrogen bond angle tends toward 180° and should preferably be above 110° ". However, if, in the case of water, the angle $\angle \text{H}\dots\text{O}$ is above 110° , then the angle HOO (used in Ref.33, which the authors follow for the q definition) cannot be larger than 35 or 40° . This, would imply that the value of q should be between $5^2=25$, from $360/(35*2)\sim 5$, and $4^2=16$, from $360/(40*2)>4$. Their choice $q=3^2$ implies, instead a $\angle \text{H}\dots\text{O}$ angle as small as 45° , in contradiction with the Footnote F4 of Ref. 46 (and all the rest of references I know).

AUTHORS: We accept the Reviewer's comment that we had misunderstanding of the definition of the angle. To better reflect the H-bonding geometry we have set $q=16$, and have re-done fitting. Thus, below Eq.13 we have changed the value of q from 9 to 16. In all related places we have altered the text to reflect the change. In particular, the changes have been made in the captions of figures and the whole subsection "Entropic cost of H-bonding with water" in Methods has been re-written.

Reviewer #3

The selection $q=6^2=36$, consistent with Ref.33, would coincide with the standard definition of a $\pm 30^\circ$ deviation from the straight HB, as used by Luzar and Chandler in <https://doi.org/10.1103/PhysRevLett.76.928>, derived from the experiments made by Teixeira, Bellisent-Funel, and Chen in <http://iopscience.iop.org/0953-8984/2/S/011>, and confirmed many times (see, e.g, <https://doi.org/10.1039/C9CP04795F> for a recent quantum calculation). A review of many works discussing the angle cut-off for the HB can be found in the section "Water hydrogen bond direction" of the web page http://www1.lsbu.ac.uk/water/water_hydrogen_bonding.html#length cured by Martin Chaplin.

I believe that the mismatch between the choice $q=3^2$ and $q=6^2$ could be due to a misunderstanding of the angle definitions, where the $\angle \text{H}\hat{\delta}\hat{\delta}\hat{\delta}\dots\text{O}$ angle and the HOO angle are confused (e.g., see the wrong use of the " $\angle \text{H}\hat{\delta}\hat{\delta}\hat{\delta}\dots\text{O}$ angle" terminology used in <https://doi.org/10.1103/PhysRevLett.76.928> when referring, instead, to the HOO angle).

Hence, the authors should discuss in more details how their results depend

on the choice $q=3^2$, even if the HB definition is beyond the scope of their study.

Did they made calculations using $q=4^2$, 5^2 , or 6^2 ?

AUTHORS: We have chosen to follow the $q=4^2=16$ choice here. Other values of q we didn't check. The text of the corresponding subsection after Eq. 15 is deleted as irrelevant.

Reviewer #3

They only write: "we have also tried a fit without fixed value of q , and found only slight deviations (about 10%) from $q = 9$ estimate". This observation should be elaborated in more details, explaining how these alternative fits were done, e.g., which parameters were free and which were fixed and how. A possible outcome could be that the physical interpretation of the parameter q is different from what they present here.

AUTHORS: It might request a special thorough study and we have removed the statement as it is too loose. Thank you for the comment, we might perform such a study in the future.

Reviewer #3

In general, the presentation is clear, although it could be improved in a few points.

AUTHORS: Thank you, we will address your advices point-by-point.

Reviewer #3

For example,

- The adverb "perfectly", in the sentence "Our model perfectly fits to the Circular Dichroism experimental data..." of the abstract, could be replaced with "Our model fits very well to ..."

AUTHORS: Accepted, changes made.

Reviewer #3

- In the introduction they mention that previous fitting methods "result in non-matching sets of fitted parameter values for Circular Dichroism (CD) and calorimetric data of the same protein". It is not clear to me is their method, instead, is able to match both CD and calorimetric data of the same protein. Can they comment in the manuscript about this point and, if possible, show it with a plot?

AUTHORS:

The research in exactly this direction is planned. Heat capacity can be calculated within our approach and is left for the future study. The conclusion about "... non-matching sets of fitted parameter values for Circular Dichroism (CD) and calorimetric data of the same protein" is one of the main results of Ref.3 (cited), not ours. Following your request we have added two sentences on p.3:

"With the approach we suggest, a procedure for the calorimetric data can be formulated as well. It would be interesting to see if it gives the similar set of fitted parameters, however, we will leave this question for future studies and concentrate here on introducing the method to extract the parameters of the water-protein interactions from the Circular Dichroism data."

Reviewer #3

- Although it is true that the anomalies of water are more evident below 4C, the statement that "in the physiologically relevant range and close to normal conditions, nothing decisive happens with water itself" should be emended, because it is not correct. Water at ambient condition has the isothermal compressibility and the isobaric heat capacity that increase for decreasing temperature, at variance with normal liquids. These anomalous behaviours of the response functions could be essential to understand the relevance of water in biological systems, because they are related to the anomalous behaviour of water's free energy (see, for example, https://doi.org/10.1007/978-3-540-78765-5_1 for a short review).

AUTHORS: We accept your comment. Our goal is to justify the use of a simpler water model. The phrase "nothing decisive happens with water itself" has been changed to "water has no phase transitions".

Reviewer #3

- The statement "the water-polypeptide interactions mostly span up to the second water layer" should be supported at least with a reference, and it should be mentioned that it is known that the effect of biological interfaces can span much more than two layers of water (see, for example, <https://dx.doi.org/10.1021/acsnano.0c02984> for models and <http://dx.doi.org/10.1103/PhysRevLett.106.158102> for experiments). Hence, the statement should be considered as a (reasonable) approximation.

AUTHORS: At the beginning of the paragraph we mention that below are the simplifications we accept true. We have changed the phrase "mostly span up to the second water layer" to "perturb only the first water layer" and have added a citation to Persson, F., Soderhjelm, P. & Halle, B. The spatial range of protein hydration. *J Chem Phys* 148, 215104 (2018).

Reviewer #3

- There are mismatches of notations and formulas between the main text and the SI, e.g., Eq.(2) in the main text and Eq.(8) in SI, or the definition of W.

AUTHORS: Typos corrected, thank you.

Reviewer #3

Furthermore, often (but not always) the Boltzmann constant is (unnecessarily) set to 1 without mentioning it.

AUTHORS: Besides the fact that the very first sentences of Results section discuss the units, we have added an explanatory footnote on p.6, saying: "We set $k_B=1$ while deriving the formulas and recover its value once we start processing the experimental data."

Reviewer #3

Also, they use the notation log instead of the more appropriate ln, for the Neperian logarithm.

AUTHORS: We accept the comment of the Reviewer and have changed log to ln throughout the text.

Reviewer #3

Furthermore, it would be helpful to use different symbols for the "=" in equations and "=" for definitions. All these notation issues make difficult to follow the discussion.

Reviewer #3

- When they introduce the reduced temperature τ , it would be better to use parenthesis to avoid misunderstandings. Also, they should clarify the advantage of using the factor $\log Q$ in its definition.

AUTHORS:

Thank you for the comment.

On p.9, above Eq.(11) we explicitly say:

"For convenience, we choose internal units of temperature as $\tilde{t}_w = T \ln Q/U$ to put the transition temperature of water-free model transition at $\tilde{t}_w = 1$."

Reviewer #3

- When describing the explicit-water model on page 5 of the main text, they should clarify better why each broken HB has only two spins described the interaction with water.

AUTHORS:

On p.6 of the main text, where we introduce the water model, we have explained why just two spins (in response to another Reviewer's request): "...spins describe water orientations around each peptide unit, two spins ($l = 1, 2$) per each broken H-bond reflect two binding sites for water...". N-H and C=O groups have only one binding site each.

Reviewer #3

-If I understand correctly, it is because the authors consider the (one-dimensional) polypeptide chain embedded into two dimensional water. In any case, apart from this, i) they should clarify the dimensionality of the model at the beginning of its definition,

AUTHORS:

We have added "one-dimensional" into the description on p.6 of SI, first sentence of "HAMILTONIAN FORMULATION OF ZIMM-BRAGG MODEL" section. Water model is 1D as well.

Reviewer #3

ii) they should mention somewhere if they expect any dependence of their results on the dimensionality of the model and why.

AUTHORS: The very first sentence of SI says:

"Zimm and Bragg suggested a one dimensional model with short-range interactions, that sets rules for assigning statistical weights to repeat units of coil (c) and helical (h) conformations."

Since we aim at introducing water-protein interactions into 1D ZB model, speaking about the dimensionality will lead us far from the goal. There is a hierarchy of models (without water), based on Potts-like approach with many-body (Δ -particle) interactions (see, eg

Ref[21]), including Zimm-Bragg and Lifson-Roig models as particular cases of nearest-neighbor ($\Delta=2$) and ($\Delta=3$) 3-particle short-range interactions. The original model is itself a particular case of Munoz-Eaton model. However, we didn't study what are the consequences of adding water beyond 1D and therefore cannot discuss these questions here.

Reviewer #3

- Is their model including any effect of the protein interface on the w-w interaction? Up to Eq.(12) I was convinced that there was no w-w interaction. However, in Eq.(12) the energy U_{ss} refers to a w-w interaction. Then, I must have missed the description of the effect. In any case, they should make a comment about it, because in other models this effect is shown to be very relevant (see, e.g., their ref. 22, 23 or <http://dx.doi.org/10.1103/PhysRevLett.103.037803>). If in their model there is no w-w interaction, then I wonder if the introduction of U_{ss} and U_{pp} , just before Eq.(12), is necessary.

AUTHORS:

Thank you for the comment. In response we have added a footnote on p.11: "To be able to correctly estimate the balance of energies in our simplified model for water, we have to take into account, that to be able to form a H-bond with N-H or C=O groups, the water-water bond has to be broken. Besides this point, we do not anyhow describe the water-water interactions in our simplified approach to water-protein interactions."

Reviewer #3

- Eq. (9) shows that the renormalized p-p energy coincides with the original U (i.e., the w-p interaction has no effect) in the limit of large q , i.e., if the entropy of water is very large. Can the authors comment on the meaning, within their model, of this limit?

AUTHORS:

Thank you for the comment. Parameter q describes the entropic cost of locating water molecule near polypeptide. Larger is q , larger is entropic cost $k_B \ln q$. It means, that in terms of free energy, the formation of water-polypeptide H-bonds is impossible (at fixed H-bonding energy). Therefore, we return to the model without solvent. In terms of math, please consult Eq.(5) (and formula A2 of Ref.25): the solvent part of partition function at infinite q 's returns a constant, and is thus not affecting the degrees of freedom of polypeptide.

Reviewer #3

- In the sentence at the beginning of page 9 they uses the expression "critical behaviour" (and later "transition"), but, in my understanding, there is no phase transition in the model. They should replace this expression with something more precise, because critical behaviour in this context would generate misunderstanding.

AUTHORS: We have changed "critical behaviour" to "conformational transitions" to clarify the meaning.

Reviewer #3

- On page 9 "[...] the energy dependence of s° at different temperatures, which becomes non-bijective for $\hat{I} < 0$ ", should be "... $\hat{I} > 0$ ".

AUTHORS: Thank you for the comment, changes made.

Reviewer #3

- On page 11, before Eq.(14) is the correct reference to Eq.(18), instead of (16)?

AUTHORS: The detailed expression showing the explicit dependence of helicity degree Eq.(14) is the Eq.(16) (Methods section). To clarify the meaning of the reference, we have re-written a sentence on p.11 to sound: "... (see Eq. 16 for the explicit expression) ..."

Reviewer #3

- On page 13, is the sentence " between the two critical points of water" referring to the liquid-gas and the (hypothesised) liquid-liquid critical point? Please, clarify.

AUTHORS: We have clarified the statement, by changing it to: "...between the two (solid-liquid and liquid-vapor) critical points of water..."

Reviewer #3

- In the first sentence on pag. 7 of the SI, "and" should be replaced with ", " .

AUTHORS: We have made the suggested changes.

Reviewer #3

In conclusion, this manuscript presents a work that is interesting and worth to be published. However, the authors should consider the points indicated in this report and they should resubmit a revised version, before the manuscript can be accepted.

AUTHORS: We are grateful to the Reviewer 3 for the constructive comments and positive opinion. We have addressed all the comments of the Reviewer and hope the revised version will be acceptable.

REVIEWERS' COMMENTS:

Reviewer #1 (Remarks to the Author):

The answers to my comments are satisfactory. The presentation of the manuscript and description of the scope of the research has significantly improved. This paper will be interesting for the broad readership of the communications chemistry.

Reviewer #2 (Remarks to the Author):

The authors have largely addressed my previous concerns, though it would be better if they put more of their explanation in the manuscript itself for the benefit of the readers instead of only in the rebuttal letter or in the supporting material. Nonetheless, I understand that there is a severe formatting constraint imposed by this journal, so I recommend publication of this revised version of the manuscript.

Reviewer #3 (Remarks to the Author):

The authors revised the manuscript in a substantial way, taking into account all my comments. I read the new version with renewed interest and found it clear.

I have a few comments that can be accounted for by the authors without further review because they are straightforward and I believe that the authors will agree with me without difficulties.

1. As an answer to one of my comments, they write on page 14 of the new version "between the two (solid-liquid and liquid-vapor) critical points of water." Of course, there is no solid-liquid critical point of water and I believe that the authors mean "between the two (solid-liquid and liquid-vapor) phase transitions of water."

2. On page 17 of the new version they write "if the water-polypeptide energy is too high, the system is denatured at any temperature. Thus, it becomes clear that opposed to the previous claims, the folding phase diagram is elliptic only for some but not all proteins". However, the general statement that proteins have an elliptic stability region of their folded state in the pressure-temperature diagram is not contradicted by their results, for at least two reasons.

First, the present work does not show the pressure-dependence of the stability region, but only the temperature-dependence at constant (ambient) pressure, and there is no reason why any protein should undergo cold denaturation at ambient pressure or any other constant pressure near ambient.

Second, the elliptic region could be centered at such a low temperature that the cold denaturation could be impossible to observe at any pressure. This is, for example, the case shown in the theoretical model presented in Fig.3a of <http://dx.doi.org/10.1016/j.molliq.2017.08.026> (the 50% curve corresponds to the stability limit), although all the stability regions in the theoretical model are elliptic and their eccentricity depends on the protein-specific parameters.

Hence, the authors should delete this sentence or modify it in a way similar to "if the water-polypeptide energy is too high, the system is denatured at any temperature. The absence of the cold denaturation discussed also in other theoretical models where the folding phase diagram is elliptic for all proteins [see, for example, <http://dx.doi.org/10.1016/j.molliq.2017.08.026>], is here a consequence of the balance between inter- and intra-molecular H-bonds energies".

3. Finally, I have a comment that, if they like, they can add to their discussion. The new Table I of

the Supplementary Info, clearly shows that the two cases with cold denaturation (c and d in the Table) differ mainly in the values of Q (hence $Q-1$, the number of accessible non-helical states for the peptides): for the case d, Q is one order of magnitude lower than for the case c. The difference between the two cases is, in my understanding, that in d the cold denaturation is a consequence of the addition of the denaturant. Hence, their model implies that the addition of denaturant largely limits the number of non-helical states for the peptides, facilitating the cold-denaturation.

I leave to the authors to accept here or to keep for future works my comment 3.

In conclusion, if, as I believe, the authors agree that my comments 1 and 2 do not change their main conclusions and results, the suggested modifications can be included in the final version without further review on my side and the paper can be accepted for publication.

Please see our point-by-point answers below.

Reviewer #1 (Remarks to the Author):

The answers to my comments are satisfactory. The presentation of the manuscript and description of the scope of the research has significantly improved. This paper will be interesting for the broad readership of the communications chemistry.

AUTHORS: We appreciate your constructive criticism which definitely improved our paper. Thank you for the final positive opinion!

Reviewer #2 (Remarks to the Author):

The authors have largely addressed my previous concerns, though it would be better if they put more of their explanation in the manuscript itself for the benefit of the readers instead of only in the rebuttal letter or in the supporting material. Nonetheless, I understand that there is a severe formatting constraint imposed by this journal, so I recommend publication of this revised version of the manuscript.

AUTHORS: We appreciate your constructive criticism which definitely improved our paper. Thank you for the final positive opinion!

Reviewer #3 (Remarks to the Author):

The authors revised the manuscript in a substantial way, taking into account all my comments. I read the new version with renewed interest and found it clear.

AUTHORS: Thank you, we worked hard to take your valuable opinion into full consideration.

I have a few comments that can be accounted for by the authors without further review because they are straightforward and I believe that the authors will agree with me without difficulties.

1. As an answer to one of my comments, they write on page 14 of the new version "between the two (solid-liquid and liquid-vapor) critical points of water." Of course, there is no solid-liquid critical point of water and I believe that the authors mean "between the two (solid-liquid and liquid-vapor) phase transitions of water."

AUTHORS: Accepted, changes made in the text of manuscript.

Reviewer #3

2. On page 17 of the new version they write "if the water-polypeptide energy is too high, the system is denatured at any temperature. Thus, it

becomes clear that opposed to the previous claims, the folding phase diagram is elliptic only for some but not all proteins". However, the general statement that proteins have an elliptic stability region of their folded state in the pressure-temperature diagram is not contradicted by their results, for at least two reasons.

First, the present work does not show the pressure-dependence of the stability region, but only the temperature-dependence at constant (ambient) pressure, and there is no reason why any protein should undergo cold denaturation at ambient pressure or any other constant pressure near ambient.

Second, the elliptic region could be centered at such a low temperature that the cold denaturation could be impossible to observe at any pressure. This is, for example, the case shown in the theoretical model presented in Fig.3a of <http://dx.doi.org/10.1016/j.molliq.2017.08.026> (the 50% curve corresponds to the stability limit), although all the stability regions in the theoretical model are elliptic and their eccentricity depends on the protein-specific parameters.

Hence, the authors should delete this sentence or modify it in a way similar to "if the water-polypeptide energy is too high, the system is denatured at any temperature. The absence of the cold denaturation discussed also in other theoretical models where the folding phase diagram is elliptic for all proteins [see, for example, <http://dx.doi.org/10.1016/j.molliq.2017.08.026>], is here a consequence of the balance between inter- and intra-molecular H-bonds energies".

AUTHORS: Reasoning above and the sentence modification accepted, changes made in the manuscript, Reference added (Ref.45).

Reviewer #3

3. Finally, I have a comment that, if they like, they can add to their discussion. The new Table I of the Supplementary Info, clearly shows that the two cases with cold denaturation (c and d in the Table) differ mainly in the values of Q (hence $Q-1$, the number of accessible non-helical states for the peptides): for the case d, Q is one order of magnitude lower than for the case c. The difference between the two cases is, in my understanding, that in d the cold denaturation is a consequence of the addition of the denaturant. Hence, their model implies that the addition of denaturant largely limits the number of non-helical states for the peptides, facilitating the cold-denaturation.

I leave to the authors to accept here or to keep for future works my comment 3.

AUTHORS: We thankfully accept the comment, but leaving it for more detailed studies. Current paper is intended to present the method and the consequences of its applications will be considered by us in the future, in line with the choice you have kindly offered.

Reviewer #3

In conclusion, if, as I believe, the authors agree that my comments 1 and 2 do not change their main conclusions and results, the suggested modifications can be included in the final version without further review on my side and the paper can be accepted for publication.

AUTHORS: We have accepted and implemented your comments 1 and 2. We appreciate your constructive criticism which definitely improved our paper. Thank you for the final positive opinion!